# Recreating Tissue Structures Representative of Teratomas In Vitro Using a Combination of 3D Cell Culture Technology and Human Embryonic Stem Cells

**DOI:** 10.3390/bioengineering9050185

**Published:** 2022-04-22

**Authors:** Alejandro Hidalgo Aguilar, Lucy Smith, Dominic Owens, Rebecca Quelch, Stefan Przyborski

**Affiliations:** 1Department of Biosciences, Durham University, Durham DH1 3LE, UK; alejandro.hidalgo-aguilar@durham.ac.uk (A.H.A.); l.a.smith2@durham.ac.uk (L.S.); dominic.owens@utoronto.ca (D.O.); rebeccaquelch@ymail.com (R.Q.); 2Reprocell Europe, NETPark, Sedgefield TS21 3FD, UK

**Keywords:** human embryonic stem cells, three-dimensional cell culture, tissue differentiation, teratoma assay, scaffold

## Abstract

In vitro studies using human embryonic stem cells (hESCs) are a valuable method to study aspects of embryogenesis, avoiding ethical issues when using embryonic materials and species dissimilarities. The xenograft teratoma assay is often traditionally used to establish pluripotency in putative PSC populations, but also has additional applications, including the study of tissue differentiation. The stem cell field has long sought an alternative due to various well-established issues with the in vivo technique, including significant protocol variability and animal usage. We have established a two-step culture method which combines PSC-derived embryoid bodies (EBs) with porous scaffolds to enhance their viability, prolonging the time these structures can be maintained, and therefore, permitting more complex, mature differentiation. Here, we have utilised human embryonic stem cell-derived EBs, demonstrating the formation of tissue rudiments of increasing complexity over time and the ability to manipulate their differentiation through the application of exogenous morphogens to achieve specific lineages. Crucially, these EB-derived tissues are highly reminiscent of xenograft teratoma samples derived from the same cell line. We believe this in vitro approach represents a reproducible, animal-free alternative to the teratoma assay, which can be used to study human tissue development.

## 1. Introduction

Understanding the events that occur during early embryogenesis and tissue differentiation continues to be a key aim in the field of human development, as well as being fundamentally important in fields such as regenerative medicine and pluripotency. Although much has been achieved and elucidated since the initial derivation of human embryonic stem cells in the 1990s [1], there is still much to be established regarding the differentiation of the cells and tissues which ultimately comprise the human body. Due to the difficulties and legalities surrounding the use of human embryos/embryonic materials [2,3], and the lack of similarity between representative animal models and human developmental events [4,5], in vitro studies employing human pluripotent stem cell (hPSC) lines continue to be a highly useful and popular surrogate for the study of human embryogenesis and tissue differentiation. This has led to the development of a wide range of protocols for the differentiation of various tissues from this invaluable cell population. With the advent of more advanced techniques, such as 3D cell culture, there are a multitude of options for altering the cellular microenvironment to achieve tissue complexity. While these protocols are useful to study certain cellular mechanisms and have yielded exciting results, variability in parameters can make it difficult to select an appropriate protocol for experimentation, and poor differentiated yields continue to hamper the transition of these promising studies beyond the bench [6,7].

Other techniques available to study human tissue differentiation include the xenograft teratoma assay. Perhaps under-utilised as an option to investigate human stem cell differentiation in the context of human embryonic development, the assay has long been considered the ‘gold standard’ method to confirm pluripotency in novel putative hPSCs [8,9,10]. The teratoma xenograft assay provides conclusive proof as to whether the stem cell population in question can form tissues from all three primary germ layers. Generally, the assay involves the implantation of the putative hPSCs into a subcutaneous or internal location in an immunodeficient mouse host. Consequently, a complex and differentiated yet benign tumour forms over the coming months. Lack of tumour formation or the absence of some germ layers may indicate assay failure and brings into question the developmental pluripotency of the population under examination. The potential of the teratoma assay as a multidisciplinary tool able to do more than simply confirm pluripotency was recognised by Müller [11] and some of these additional applications have started to be fulfilled. It has been indicated that the assay is crucial for determining malignant potential in hPSCs [12,13]; various studies into topics such as the nature of pluripotency [14], human development [15,16] and neoplasia [17,18] have used the assay as a key technique. Indeed, multiple investigations have shown the ability to directly modulate the nature and composition of the tissues within the tumour, by altering the location [19,20], cell number [21] or components of the implantation [22], allowing for some degree of differentiation control. This could be useful if the assay were to be further pursued and employed to study human tissue development.

However, the xenograft teratoma assay itself is not without issue. The principle of the ‘3Rs’ in animal research was originally proposed by Russell and Burch in 1959 [23], but with recent and continual developments in new technologies, techniques and methods, we are now in a position to completely rethink the use of animals in scientific research, with a view to reducing the numbers required, refining experiments or replacing them with more accurate and predictive animal-free alternatives [24,25]. The use of large numbers of mice for human developmental studies is becoming increasingly difficult to justify, and incomplete assay outcomes may mean a waste of valuable time, resources and money and a possible need to repeat, only increasing the animal burden of experiments. Furthermore, not only is the use of animals and the associated care required during the assay highly expensive, the assay itself is laborious and requires specific animal handling training and the acquisition of appropriate government licencing. Mouse host tissue is problematic in downstream applications, with the potential to confound results due to the difficulties in completely removing this from samples. The ability to influence differentiation by altering protocol parameters is a potential strength of the assay for the study of human tissue differentiation, but variability in protocols is a particularly serious issue in the assessment of pluripotency, leading to the inability to compare the potency of differing hPSC lines due to a lack of assay standardisation [8,9,11,22]. For many years, the stem cell field has been searching for a more reliable alternative to assess the pluripotency of newly derived hPSC lineages. Together, this presents an opportunity to develop a novel model which can be used for both the study of tissue formation and the assessment of developmental potential.

Previously, we have shown the ability to prolong the viability and enhance the differentiation of EBs through seeding and maintenance on a porous, polystyrene membrane, resulting in the formation of recognisable tissues from the three germ layers [26]. We previously reported on EBs derived from human embryonal carcinoma cells as a proof of concept and more extensively on mouse embryonic stem cells to further explore the capabilities of the novel in vitro model. In this study, we further explore and expand on the use of human embryonic stem cell-derived EBs, using a similar methodology to extend their viability and differentiation. Recognising the importance of time during tissue development, we show how extending culture viability to allow for long-term differentiation significantly increases the variety and complexity of the rudimentary tissues which form, with examples from all three germ layers observed. In addition, we demonstrate the ability to manipulate tissue differentiation towards specific lineages, allowing for a more detailed study of developmental events. The tissues formed using this in vitro approach were comparable to xenograft teratoma tissues derived using the same hPSC line, therefore highlighting the potential of this novel method to be used as a reproducible and configurable replacement for the animal-based assay. 

## 2. Materials and Methods

### 2.1. H9 Routine Culture

Human embryonic stem cell line H9 (hESCs, WiCell, Madison, WI, USA) was maintained in feeder-free conditions in 6-well plates (Greiner-Bio One, Stonehouse, UK), coated with Matrigel hESC qualified matrix (Corning, Flintshire, UK) and prepared and diluted according to the manufacturer’s instructions. Cells were cultured in mTESR Plus medium (Stem Cell Technologies, Cambridge, UK) and prepared according to the manufacturer’s instructions, with daily media changes and maintenance in 37 °C 5% CO_2_, 95% air. On reaching 70–80% confluency, cells were passaged via an enzyme-free method using ReLeSR (Stem Cell Technologies, Cambridge, UK) to detach and split into fresh plates at a ratio of 1:6.

### 2.2. Formation, Maintenance and Differentiation of H9 Embryoid Bodies

Aggrewell^TM^ 800 plates (Stem Cell Technologies) were used to create embryoid bodies of a uniform size. Plates were prepared according to the manufacturer’s instructions prior to use, by washing in culture medium and centrifuging at a high speed to dislodge any air bubbles in the base of the microwells. Prior to EB formation, Rho kinase inhibitor (ROCK inhibitor) Y27632 (Tocris Biosciences, Abingdon, UK) was added to wells containing 70–80% confluent H9 cells for 60 min at a final concentration of 10 μM to promote single-cell survival. ROCK inhibitor was also added to media to be used during counting and EB formation at the same concentration. Following incubation in ROCK inhibitor, H9 cells were detached as described using ReLeSR, gently pipetted to form a single cell suspension and counted using the Trypan Blue exclusion assay to obtain accurate viable cell numbers. Cells were resuspended in differentiation medium, consisting of Knock Out DMEM (Fisher Scientific UK, Loughborough, UK), 20% Knock Out Serum Replacement (Fisher Scientific UK), 1 mM L-glutamine (Fisher Scientific UK), 0.1 mM non-essential amino acids (Fisher Scientific UK), 0.1 μM β-mercaptoethanol (Fisher Scientific UK) and 10 μM Y27632, and seeded at a density of 4.5 × 10^6^ cells per well to form 300 EBs of approximately 15,000 cells. EBs were formed over 24 h by incubating the plate at 37 °C 5% CO_2_, 95% air, before transferring to suspension culture in sterile non-treated 6-well plates (Corning, Flintshire UK)

To differentiate towards the ectoderm lineage, a combination of protein and small molecules were used as outlined in the International Stem Cell Initiative Study 2018 were used (see Table 1). Dorsomorphin was substituted with LDN193189. EBs were exposed to morphogens only during the Aggrewell^TM^ and suspension culture stages.

### 2.3. Preparation of Alvetex^®^ Membranes and Seeding/Maintenance of EBs

Well inserts containing Alvetex^®^ Polaris polystyrene membranes (Reprocell Europe, Glasgow, UK), which consist of 3 µm and are not commercially available, washed in 70% ethanol overnight, before being rinsed twice with sterile phosphate buffered saline solution (PBS). Inserts were then coated using Matrigel Growth Factor Reduced Basement Membrane Matrix (Corning) to facilitate EB attachment before being placed in 6-well plates with 5 mL of the appropriate culture medium in the well consisting of either Knockout DMEM, 20% Knockout Serum Replacement (Fisher Scientific UK), 1 mM L-glutamine, 1% non-essential amino acids and 0.1 μM β-mercaptoethanol or high glucose DMEM (Fisher Scientific), 10% Foetal Bovine Serum (FBS, Fisher Scientific UK), 2 mM L-glutamine, 55 µM β-mercaptoethanol, 0.1 mM nonessential amino acids and 100 U/mL Penicillin/Streptomycin (Fisher Scientific UK).

Once PSC-derived EBs had formed, they were transferred to the prepared Alvetex^®^ inserts, with around 50–100 EBs placed on each polystyrene insert and allowed to attach overnight at 37 °C, 5% CO_2_ and 95% air. The following day wells were topped up to 10 mL total media. EBs were then maintained with regular media changes for a defined timescale, before fixation in 4% paraformaldehyde (PFA, Fisher Scientific UK) at 4 °C overnight.

### 2.4. Sample Processing and Sectioning

Following fixation, samples were washed twice in PBS and dehydrated through a series of ethanols. Samples were dehydrated in 30% and 50% ethanol before staining in 0.1% crystal violet in 70% ethanol to improve visualisation of 3D structures during processing, embedding and sectioning. Samples were then further dehydrated through 80%, 90%, 95% and 100% ethanols, before incubation in Histoclear I for 20 min (Scientific Laboratory Supplies, SLS, Nottingham, UK), and subsequently transferred to a 50:50 solution of Histoclear I and paraffin wax (Fisher Scientific UK), with incubation at 60 °C for at least 30 min. Samples were then transferred to 100% molten paraffin wax and incubated at 60 °C for a further 60 min. Inserts were embedded in paraffin wax using plastic embedding moulds and orientated appropriately to allow for longitudinal and transverse sectioning relative to the insert. Wax blocks were sectioned at 6 µm using a Leica RM2125 RT Microtome and mounted onto SuperFrost™-charged microscope slides (Fisher Scientific UK).

### 2.5. Haematoxylin and Eosin Staining

In order to assess structures present, slides were stained using haematoxylin and eosin (H&E). Slides were deparaffinised in Histoclear I and rehydrated through a series of ethanols, finishing in distilled water. Slides were stained in Mayer’s haematoxylin (Sigma Aldrich, Dorset, UK) for 5 min, before washing in distilled water for 30 s. Slides were incubated in alkaline ethanol for 30 s to blue the nuclei, before incubation in 95% ethanol for 30 s. Slides were finally stained in eosin dissolved in 95% ethanol for 1 min, before dehydration through 95% and 100% ethanols. Slides were dehydrated again in 100% ethanol and cleared in Histoclear I for 5 min each before mounting using glass coverslips and Omnimount (SLS, Atlanta, GA, USA).

### 2.6. Immunohistochemical Staining

Slides were deparaffinised in Histoclear I for 15 min before being rehydrated through a series of ethanols, finishing in PBS. Antigen retrieval was performed by incubating the slides in citrate buffer (pH 6) for 20 min at 95 °C. Once slides had cooled, samples were incubated in blocking buffer consisting of 20% normal calf serum in 0.4% Triton-X−100 in PBS for 1 h at room temperature. Primary antibodies diluted in blocking buffer were then added to the slides at the appropriate concentrations (see Table 2) and incubated overnight at 4 °C. The next day, slides were washed three times in PBS before fluorescently conjugated secondary antibodies and the nuclear stain Hoechst were added to the slides diluted in blocking buffer (see Table 3). Slides were incubated at room temperature for 1 h before washing three times in PBS and being mounted with glass coverslips using Vectashield™ Antifade Mounting Medium (Vector Labs, Peterborough, UK), with nail varnish to seal the edges. Slides were stored at 4 °C until imaging using a Zeiss LSM 880 confocal with Airyscan.

### 2.7. Teratoma Xenograft Assay

H9 hESCs were routinely maintained in feeder free conditions as described above. On the day the assay was commenced, cells were detached using 0.25% trypsin/2 mM EDTA (Fisher Scientific) and counted using the Trypan Blue Exclusion Assay. The cell suspension was adjusted to a density of 5 × 10^5^ cells per 100 μL, and subsequently, 100 μL of cell suspension was combined with 100 μL of Growth Factor Reduced Matrigel matrix (Corning, Fisher Scientific UK). This was loaded into 1 mL syringes with 21G needles which had been cooled prior to use. Loaded syringes were kept covered and maintained on ice during transport to the animal facility. The animal hosts were adult male nude (nu/nu) mice, and each host was injected subcutaneously with one loaded syringe into each flank, with three mice injected for each condition giving a potential for six total tumours per condition. Mice were examined regularly during the assay and when tumour formation was noted, the growth was measured at least once a week until a size of 1 cm^2^ was reached, at which point the animal was sacrificed and the tumour surgically removed. All procedures were completed under licence and permission according to the guidelines of the Home Office, United Kingdom.

## 3. Results

### 3.1. Teratomas Formed from Human Pluripotent Stem Cells Are Composed of Diverse Tissue Derivatives of the Three Embryonic Germ Layers

Currently, the xenograft teratoma assay is the only available in vivo method to assess the developmental potency of human pluripotent stem cells, due to the inability to perform other more stringent methods which use chimeric organisms, such as germline transmission or tetraploid complementation. The teratoma assay demonstrates the ability of a hPSC line to form tissue structures from all three primary germ layers, with Figure 1 showing a representative example of the variety and complexity of structures which may form within a teratoma derived from a human embryonic stem cell population. These xenografts consist of a diverse array of differentiated cell types, which are partially organised into recognisable rudimentary tissues (A), whose identity can be further confirmed using histological and antibody based staining methods. Examples from each germ layer are as follows: ectoderm-neuroepithelium and squamous epithelium (D and F), endoderm-general epithelial and glandular structures (B, C, E and H) and mesoderm-smooth muscle and cartilage (G and I).

### 3.2. A Novel In Vitro Culture Method to Improve the Cellular Microenvironment and Enhance the Differentiation of Human Pluripotent Stem Cell-Derived Tissues

For many years, embryoid bodies have been a valuable model to study basic aspects of human embryogenesis and as an initial step for several differentiation protocols. Nevertheless, as the EB increases in size, differential oxygen and nutrients gradients are formed across the structure, which affects the differentiation, viability and proliferation of the PSCs and the extent to which EBs can be utilised (Figure 2A). We have previously developed a novel in vitro culture method in order to extend the viability of these structures, whereby PSC-derived spherical EBs are seeded and maintained as flattened 3D “discs” of cells on the surface of a porous polystyrene scaffold (Figure 2A). Here, we apply this methodology to human PSC and derived EBs, as shown in Figure 2B, where the gross morphology of EBs, maintained either in suspension or on the surface of the porous scaffold for up to 35 days, was compared. As shown in Figure 2B, EBs in suspension culture exhibited a much lower level of differentiation compared to the EBs grown on porous substrates and were smaller overall in size. The EBs were comprised of limited tissue structures, mainly consisting of simple connective tissue, indicating mesodermal differentiation, with some areas displaying neuroepithelial differentiation, indicating the presence of ectodermal derivatives. A few simple epithelial structures could be observed, but large areas of cell death were also found in the centres of the PSC aggregates. By contrast, EBs that were maintained on the porous scaffold membranes showed the formation of a variety of recognisable and more complex tissue structures from all three primary germ layers. These structures included extensive ectoderm-derived neuroepithelia, mesodermal-derived tissue such as rudimentary cartilage and endoderm-derived epithelial structures with evidence of organisation and polarisation.

### 3.3. Improvement of Viability Allows for Increased Time in Culture, Resulting in the Formation of More Complex and Mature Tissue Structures from hPSC-Derived EBs

We have previously demonstrated how maintenance of EBs on porous membranes enhanced their viability and reduced evidence of a necrotic core [26]. In order to assess the impact of increasing time in culture on the identity and complexity of the tissues and structures within the EBs maintained on the scaffold, EBs derived from the human ES cell line H9 were seeded onto the porous membranes and maintained in culture for up to 21, 28 and 35 days. Low magnification imaging of differentiating cells showed that maintaining 3D cultures for a longer period of time notably increased the size, maturity and complexity of the tissue structures from all three germ layers, as can be seen in representative images in Figure 3A. Neural tissue, in the form of neural tubes and neural rosettes, was frequently observed in all of the 3D cultures, with the subsequent increase in area notable at day 35. Many tissues were surrounded by large areas of mesoderm-derived connective tissues, which also increased in their diversity over time, showing varied areas rich in ECM proteins and mesenchymal cell populations. The organisation and diversity of epithelial structures was also enhanced by increasing the time in culture, with a range of epithelial cell morphologies and structures noted.

Immunostaining was performed to confirm the identity of the tissue structures observed in the 3D cultures (Figure 3B), using a number of protein markers generally correlating to each germ layer. Neural tissue stained positive for the early neural marker nestin and the pan neural marker Class III β tubulin, indicating multiple neural cell identities within the structures. Endoderm-derived epithelial structures were identified by the expression of the epithelial cell junctional protein, E-cadherin, positive staining of which indicated the formation of an organised layer. The presence of mesodermal tissue derivatives was confirmed by the detection of vimentin, a general marker for mesenchymal cells and fibronectin, an extracellular matrix component.

### 3.4. Application of Exogenous Morphogens to Direct EB Differentiation Results in the Formation of Tissues from Specific Lineages

To demonstrate that it was possible to direct the differentiation of tissue structures towards a given germ layer, H9 EBs were stimulated with a combination of morphogens at the Aggrewell^TM^ and suspension stages of the method in order to form tissues from the ectoderm. The morphogen combination was taken from the International Stem Cell Initiative study [12] with a minor modification and was as follows: 10 μM SB431542, 0.25 μM LDN193189 and 100 ng/mL bFGF. Following treatment, EBs were seeded onto the porous scaffold membranes as previously described. As shown in the representative images in Figure 4A, large areas of neural differentiation surrounded by mesoderm-derived connective tissue were clearly visible within the 3D cultures. Neural tissue was confirmed through a small amount of nestin-positive cells and significant areas of Class III β tubulin staining (Figure 4B), indicating mature neural tissue. Vimentin and fibronectin expression indicated the presence of connective tissue. Overall, mesodermal differentiation was limited to connective tissues with no evidence of more specialist structures such as cartilaginous proformas. The lack of visible epithelial layers or structures combined with an absence of positive staining for the junctional protein E-cadherin suggested the inhibition of endodermal differentiation within the EB.

### 3.5. In Vitro Tissues Derived from hESC Are Highly Similar to In Vivo Xenograft Teratomas Derived from the Same Cell Line

Spontaneously differentiated H9 EBs maintained on porous scaffolds were capable of developing complex, differentiated structures from each germ layer, which were highly similar to the differentiated tissue structures found in teratoma xenografts formed from the same cell population (Figure 5B). This similarity demonstrates the ability of our novel in vitro method to form complex and recognisable tissue structures, which could be used to assess the pluripotency of hPSCs and study human tissue development in future applications. Furthermore, our in vitro method offers a wide range of advantages over the in vivo xenograft assay, including the analysis of multiple samples at the same time, compatibility with multiple analytical techniques, reproducibility and reduction of costs (Figure 5A).

## 4. Discussion

Human embryogenesis, including organogenesis and tissue differentiation, is a complex process which involves an organised series of events, including cell proliferation, division, migration and differentiation, usually through the activation of different signalling pathways in a spatial-temporal manner. Currently, a range of both in vitro and in vivo approaches are employed to investigate these processes, each with its own advantages and disadvantages regarding the methodology and information that can be acquired. Nevertheless, the formation of complex, recognisable and mature differentiated tissues from all three germ layers using hPSCs can only be achieved with the teratoma assay (see Figure 1). While the teratoma assay can be used to study human development, the technique has a number of limitations already established by the stem cell field, including the use of animals, significant protocol variation between laboratories resulting in the inability to compare data and a highly labour-intensive methodology. As such, its use in these fundamental human tissue development studies is problematic. Previously, we have shown the ability of our in vitro model to recapitulate aspects of the tissue complexity and maturity found in teratoma xenografts using EBs derived from embryonal carcinoma cells and murine embryonic stem cells [26]. In this study, we further explore the capabilities of the model using hESC-derived EBs.

Embryoid bodies have been widely used as in vitro models to study embryogenesis and tissue differentiation and to assess the cellular pluripotency of PSCs, due to their ability to recapitulate early events of embryonic development [27,28,29]. However, increasing EB size results in the formation of various gradients across the developing structure, crucially oxygen and nutrient gradients, which directly impact on the viability and differentiation occurring within the cellular aggregate (see Figure 2A(i)) [30]. The formation of a central core of cell death [31,32] renders EBs an unsuitable model for the longer-term studies necessary to enable the formation and organisation of complex tissues. Previously, we have reported the ability to enable a higher degree of tissue differentiation and complexity in EBs through seeding onto a porous scaffold, with subsequent structures resembling the tissue morphologies observed in teratoma xenografts [26]. Similar approaches have been explored to improve the longevity, differentiation and organisation of neurospheres by creating a more physiological in vitro microenvironment through the reduction of diffusion distances to prevent the formation of unwanted/uncontrollable gradients [33,34,35]. Using the two-step culture method, initial results revealed differentiated tissues from all three primary germ layers in both hESC line H9-derived EBs maintained in suspension and those maintained on porous scaffolds for up to 35 days. Critically, histological analysis of EBs maintained in suspension culture showed the formation of a very limited number of tissue-like structures, with EBs retaining a relatively small size. This low degree of differentiation has been previously reported in studies examining EB morphology [36,37,38]. In contrast, the culture of EBs on porous scaffold membranes resulted in the development of various cell types derived from all three germ layers organised into complex, recognisable tissue structures, with the low magnification image highlighting the diversity within the resultant EB-derived structure (Figure 2B). Large areas of neural components including neuroepithelium and neural rosettes could be observed in EBs maintained on porous scaffolds; this may be simply due to the fact that the default pathway of spontaneous differentiation for hESCs is neuroectoderm [39] or due to inherent variability in the lineage preferences of hPSCs [40]. In addition to this notable ectodermal component, clear examples of the presence of endodermal and mesodermal derivatives were observed.

Time plays a crucial role during human development, regulating the delicate balance between proliferation, gene expression and differentiation, and influencing the growth assembly of developing tissues, including their cellular composition, size and maturity [41]. The importance of time in developmental and differentiation focussed studies cannot be overstated, yet current cell culture technologies can be limiting in the time that cellular populations and constructs can be maintained, leading to the formation of immature tissues or resulting in premature cell death due to an unfavourable cellular microenvironment [42,43]. In this study, we were able to determine the impact of prolonged maintenance on the differentiation of EBs, observing an enhanced complexity and diversity of tissue structures. Analysis of EBs maintained on porous scaffold membranes after 21 days, and the subsequent time points, revealed an increase in size, heterogeneity and maturity of the tissue structures derived from the three germ layers, with the absence of any observable areas of cell death within these long-term cultures. We conclude that this 3D in vitro model is an attractive method to recapitulate in vivo developmental stages due to the ability to undertake longer term studies, which permit the formation of more mature tissues.

Directing the differentiation of hPSCs is incredibly useful to study specific germ layers, differentiation pathways and developmental events [44]. In an attempt to direct the tissue differentiation within the EB-derived structures towards a specific germ layer, we employed a basic differentiation protocol involving the single application of exogenous morphogens during the EB formation stages, using compounds which are known to influence cellular differentiation along the specified lineage. While the results showed a certain degree of variability as the tissues produced were not solely derived from the germ layer of interest, which was expected due to the simple nature of the protocol, a noticeable shift in the tissue structures produced was observed. A general increase in tissues corresponding to the ectoderm germ layer was seen, as highlighted in the images in Figure 4, which show large amounts of neural tissues staining positive for the mature neural marker Class III β tubulin. The absence of E cadherin staining within the structure suggests that endodermal differentiation was inhibited in the structures, but a small amount of mesodermal differentiation was observed. This straightforward method demonstrates the ability to manipulate the differentiation of the tissue structures in quite a basic manner; future experiments could involve the application of a series of morphogen combinations to the 3D in vitro model at specified times and relevant concentrations in order to recapitulate specific developmental events and further create specific tissue structures.

Despite being considered the “gold standard” method for assessing the functional pluripotency of hPSCs, the teratoma assay has low consistency in both the methodology used to perform the assay and the reporting of results. This has led to the search for alternatives, as well as suggestions on how to boost the reproducibility, reliability and status of the teratoma assay, which can provide a wealth of information beyond simple pluripotency assessment [8,9,11,22]. This in vitro approach is capable of forming a wide range of complex and mature tissue structures with evidence of differentiation of all three primary germ layers using the same cell populations, fulfilling the single critical success criterion of the xenograft teratoma assay [10]. Moreover, this in vitro model offers numerous advantages (see Figure 5) over the in vivo xenograft assay, including reproducibility, reduction of costs, capacity to increase sample number using the same cell population and low maintenance requirements compared to studies involving the use of animals. A wide range of analytical techniques are compatible with this in vitro model, allowing for a more in-depth investigation into pluripotency, cellular differentiation and tissue morphogenesis.

## 5. Conclusions

We have developed a novel in vitro method to enhance the viability, differentiation and maturation of PSC derivatives into tissue structures and have explored the capabilities of this technique using human embryonic stem cell-derived EBs. We have demonstrated that the methodology has the capacity to successfully support the complex differentiation of these structures over long culture periods and can be used to form differentiated tissues of the desired germ layer. The tissue structures within the in vitro model are derived from all three primary germ layers and are highly similar to those structures found in in vivo xenograft teratomas. In this study, we have developed a reproducible, animal-free alternative to the teratoma xenograft assay, which has a number of advantages over the in vivo method, including controllability, the potential for high throughput studies and reduced costs. This novel technique can be used in a variety of applications, including the detailed study of developmental events and tissue morphogenesis and the assessment of pluripotency of novel human PSC populations.

## Figures and Tables

**Figure 1 bioengineering-09-00185-f001:**
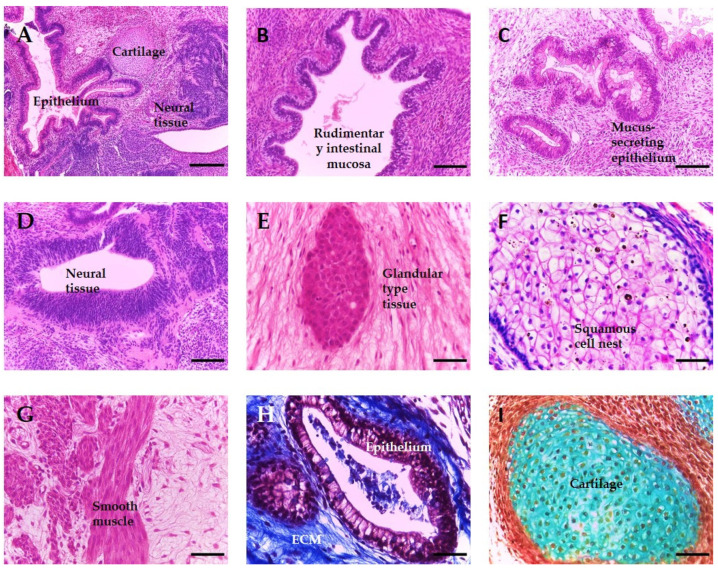
Representative identifiable tissue structures in teratoma xenografts derived from human pluripotent stem cells. When transplanted into an immunodeficient mouse, human pluripotent stem cells, such as embryonic and induced pluripotent stem cells, form teratomas xenografts, which contain a diversity of complex differentiated tissue structures. The assay is normally used for assessing the pluripotency of cells and the presence of tissue derived from all three germ layers is indicative of this. Images are haematoxylin and eosin stained unless otherwise stated. (**A**) Section of a teratoma showing derivatives of all three germ layers, including cartilage (mesoderm), epithelium (endoderm) and neural tissue (ectoderm). (**B**) Rudimentary intestinal mucosa. (**C**) Mucus-secreting cells. (**D**) Neural tissue in the form of neural rosettes. (**E**) Glandular type tissue. (**F**) Nest of immature squamous cells. (**G**) Smooth muscle tissue. (**H**) Epithelia surrounded by ECM (blue) stained using Masson’s Trichrome stain. (**I**) Cartilage (pale blue) stained using Weighert’s stain. Scale bars (**A**) 200 µm, (**B**,**C**) 100 µm and (**D**–**I**) 50 µm.

**Figure 2 bioengineering-09-00185-f002:**
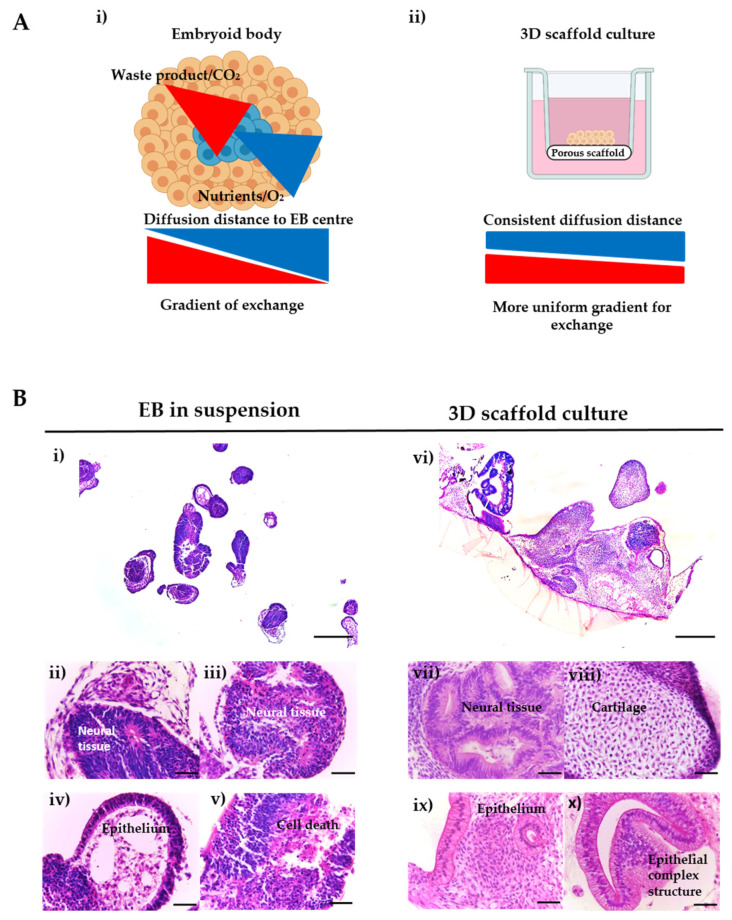
Comparison of the morphology and structures of EBs and 3D scaffold cultures derived from human embryonic stem cells. (**A**) Schematic and rationale underpinning the development of new 3D system. The availability of oxygen, nutrients and other molecules decreases towards the core of the EB, whereas CO₂ and metabolic waste accumulates at the deepest layers of the aggregate (i). EB flattening occurs when the structure is seeded on the Alvetex^®^ membranes. As a result, the diffusion distance is reduced, improving the viability and differentiation of human ES cells (ii). (**B**) Sections of H9 cell-derived EBs cultured as suspended spheroids (i–v) or maintained on porous substrates for up to 35 days (vi–x). H&E images show some degree of differentiation (i) including neural rosettes (ii–iii), simple epithelium and connective tissue (iv), as well as tissue degradation and cell death (v). By contrast, EBs cultured on porous membranes formed complex tissue structures (vi), such as neural tissue (vii), cartilage (viii), lined epithelium and possible gland structure (ix) and complex epithelial structure (x). Representative structures shown. Scale bars: (**B**(i,vi)) 500 μm; all others, 50 μm. Figure 2A was created with Biorender.com.

**Figure 3 bioengineering-09-00185-f003:**
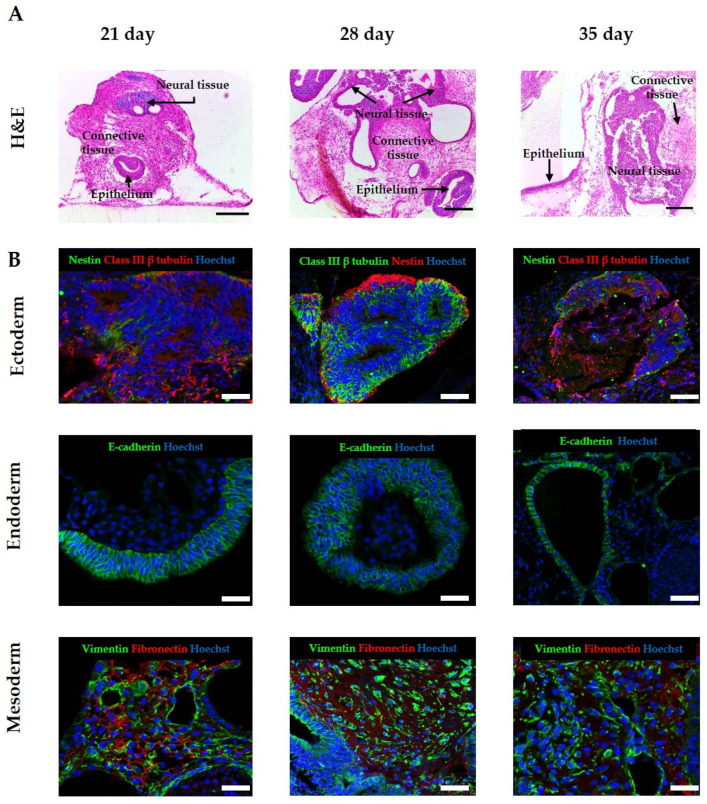
Formation of teratoma tissues derived from human ES cells in vitro: H9 embryoid bodies were seeded on porous membranes and cultured for specified times. Samples were sectioned in transverse planes to analyse the structures within the tissue discs. (**A**) H&E images reveal that at 21 days, complex tissues are present including neural structures, epithelial structures and connective tissue. Prolonged culture of human embryoid bodies on the surface of the porous membrane results in the maintenance and increased complexity of these embryonic tissue structures. Multiple different cellular phenotypes can be observed, such as organised epithelia, lumen structures, large areas of connective tissue and neuroepithelial and neural structures. (**B**) To further validate the tissue derived from the three germ layers present within the model, immunohistochemical analysis was performed using a number of germ layer markers. Positive staining of neural markers Nestin and Class III β tubulin confirmed the presence of neural differentiation. Vimentin and fibronectin staining indicated the presence of mesenchymal cell populations and connective tissue, respectively. Epithelial structures were positive for E-cadherin. Representative structures shown. Scale bars: (**A**) 200 μm, (**B**) 100 μm.

**Figure 4 bioengineering-09-00185-f004:**
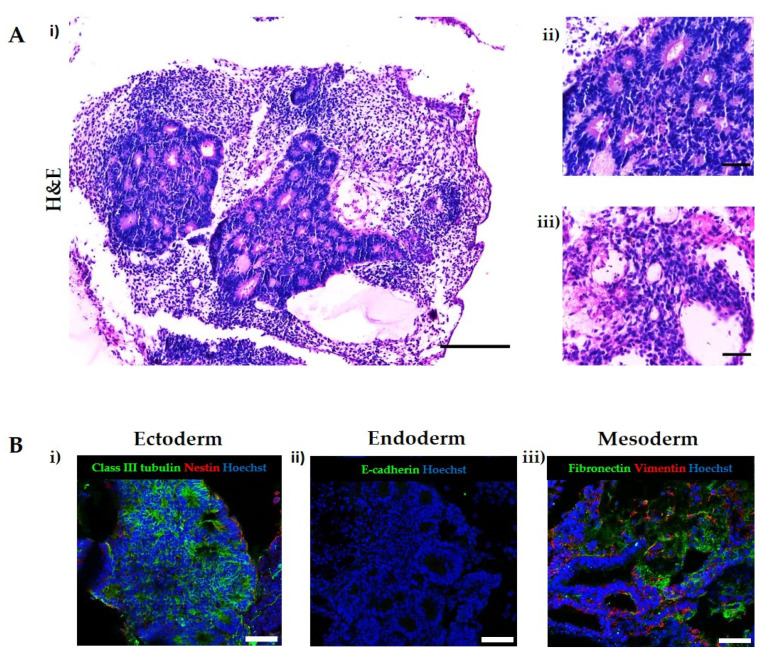
Directed differentiation of teratoma tissues derived from differentiating human ES cells for 35 days. To induce ectodermal differentiation, H9 human ES cells were treated with SMAD inhibitors, SB 431245 and LDN 193189, and FGF during EB formation and suspension stages before seeding onto porous membranes to form tissue discs. H&E reveals large areas of neural differentiation (**A**(i)) in the form of neural rosettes (**A**(ii)) surrounded by connective tissue (**A**(iii)). Positive staining for Class III β tubulin and nestin confirmed the presence of neural differentiation (**B**(i)). There are no areas of endodermal differentiation due to the lack of E-cadherin staining (**B**(ii)). The presence of connective tissue is confirmed by vimentin and fibronectin staining (**B**(iii)). Representative structures shown. Scale bars: 200 and 50 μm (H&E), 100 μm (Immunohistochemical).

**Figure 5 bioengineering-09-00185-f005:**
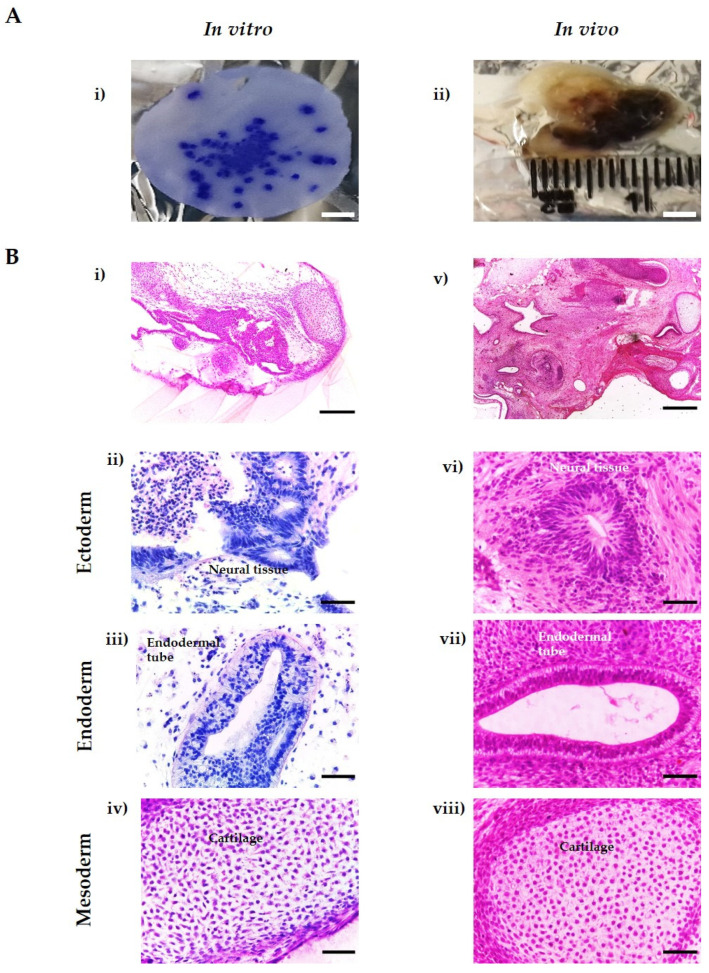
Features of in vivo and in vitro methods for producing teratoma tissues derived from human PSCs. The in vitro system offers distinct advantages compared to the in vivo method, such as capacity for high throughput studies, ability to direct the differentiation of the cells by the addition of exogenous stimuli (e.g., morphogens), improved reproducibility and reduced associated costs. From a single porous scaffold, multiple independent 3D teratoma models can be produced from the same stem cell population (**A**(i)) compared to the in vivo method (**A**(ii)). Histological analysis from H9 embryoid bodies cultured on porous membranes for 35 days reveals varied cell morphologies and complex cellular organisation (**B**(i)). H&E staining shows the presence of complex structures from the three layers such as neural rosettes (**B**(ii)), connective tissue (**B**(iii)) and polarised epithelial tissues (**B**(iv)). These structures are highly reminiscent of those seen in the xenograft teratoma tissue (**B**(v–viii)). Representative structures shown. Scale bars: 0.5 cm (**A**(i,ii)), 200 µm (**B**(i,v)) and 50 µm (all the others).

**Table 1 bioengineering-09-00185-t001:** Morphogens used to direct the differentiation of EBs.

Morphogen	Supplier	Final Concentration
SB431542	Peprotech, London, UK	10 µM
LDN193189	Peprotech	0.25 µM
Basic Fibroblast Growth Factor (bFGF)	Peprotech	100 ng/mL

**Table 2 bioengineering-09-00185-t002:** Primary antibodies used to establish specific tissue identities.

Antibody	Species	Dilution	Product Code	Supplier
Classs III β tubulin	Rabbit	1:100	ab18207	Abcam UK
E-cadherin	Mouse	1:200	610,181	BD Biosciences UK
Nestin	Mouse	1:200	ab22035	Abcam UK
Vimentin	Mouse	1:100	sc6260	Santa Cruz Biotechnology UK
Fibronectin	Rabbit	1:100	ab17808	Abcam UK

**Table 3 bioengineering-09-00185-t003:** Secondary antibodies and nuclear stains used for immunohistochemical staining.

Antibody	Dilution	Product Code	Supplier
Goat Anti-Mouse Alexa Fluor 594	1:1000	A11005	Invitrogen, Fisher Scientific UK
Goat Anti-Rabbit Alexa Fluor 488	1:1000	A11034	Invitrogen, Fisher Scientific UK
Donkey Anti-Rabbit Alexa Fluor 594	1:1000	A21207	Invitrogen, Fisher Scientific UK
Donkey Anti-Mouse Alexa Fluor 488	1:1000	A21202	Invitrogen, Fisher Scientific UK
Hoechst 33342	1:10,000	H3570	Fisher Scientific UK

## Data Availability

The original data presented in this study are included in the article. Further inquiries can be directed to the corresponding author.

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
