# Peer review of "Recreating Tissue Structures Representative of Teratomas In Vitro Using a Combination of 3D Cell Culture Technology and Human Embryonic Stem Cells"

_bioengineering, 2022, doi:10.3390/bioengineering9050185_

Round 1
Reviewer 1 Report
In this manuscript, the authors demonstrated a novel 3D in vitro model for teratoma assay using embryoid bodies derived from hESCs (hESC-EBs), which is reproducible and could be an alternative option to traditional teratoma xenograft assay. All three germ layers derived from hESC-EBs were well presented via histology staining. The advantages and potential of application of this 3D model were also well discussed. Please see below for some comments:
- For the viability of hESC-EBs, is there any necrotic core in those EBs?
- The histology comparison between in vitro derived tissue from hESC-EBs and in vivo xenograft teratomas was presented, what about the gene and protein expression level?
Author Response
1. For the viability of hESC-EBs, is there any necrotic core in those EBs?
In our previous paper “Using Advanced Cell Culture Techniques to Differentiate Pluripotent Stem Cells and Recreate Tissue Structures Representative of Teratoma Xenografts”, we assessed cell death using a Caspase 3/7 dye and TUNEL staining. Our results showed a reduced staining for Caspase 3/7 and TUNEL when EBs were cultured on the porous membrane compared to those maintained as spheroids in suspension. This data indicates an improved cell viability due to the reduction of diffusion distances as the shape of the spherical EB transforms and flattens. We have made this clear in the current manuscript and have added an additional sentence ‘We have previously demonstrated how maintenance of EBs on porous membranes enhanced their viability and reduced evidence of a necrotic core [26].’ – lines 328-9.
2. The histology comparison between in vitro derived tissue from hESC-EBs and in vivo xenograft teratomas was presented, what about the gene and protein expression level?
Our aim in this study was to demonstrate that our novel in vitro system has the capacity to provide a proper microenvironment for the formation of complex tissue structures from human embryonic stem cells comparable to xenograft teratomas tissues derived from the same lineage in vivo. The manuscript reports that this has been achieved using histological and immuno-staining techniques. We focused on characterizing the structure and anatomy of the tissues formed and comparing between the in vitro and in vivo methods. We acknowledge the point raised by the Reviewer, but it was beyond the scope of the current study to examine gene and protein levels. Moreover, precise biomarker expression levels may be difficult to compare given the complexity of the different cell types within these differentiated tissues.
Reviewer 2 Report
The study design combining in exo vivo and in vitro experiments is conceptualised well and described in great detail. I think the authors also did a great job in clearly presenting the results and carefully interpreting them. This article can be accepted for publication after some changes.
1. Human pluripotent stem cells should be “human embryonic stem cells”, please correct that. (Line 9)
2. Please provide the phenotype characterization data for ESC in this study.
3. Please provide the country name “Corning” (line 139)
4. Please provide the country name “SLS“ (line 198)
5. Please provide the country in Table II (page 6)
6. Please provide the cell viability assay in “Figure 3” after 21, 28, and 35 days of incubation.
Author Response
- Human pluripotent stem cells should be “human embryonic stem cells”, please correct that. (Line 9)
Corrected, please see manuscript.
- Please provide the phenotype characterization data for ESC in this study.
We have provided evidence that the embryonic stem cells are pluripotent (see Figure 1 – teratoma formation demonstrating formation of the three germ layers). In addition, we can include a Supplementary Figure showing the expression of the expression of pluripotency-associated markers using immunocytochemistry and flow cytometry. We have uploaded this onto the website (please see attached document) and is available if required.
- Please provide the country name “Corning” (line 141)
Provided, please see corrected version of the manuscript.
- Please provide the country name “SLS“ (line 200)
Provided, please see corrected version of the manuscript.
- Please provide the country in Table II (page 6)
Provided, please see corrected version of the manuscript.
- Please provide the cell viability assay in “Figure 3” after 21, 28, and 35 days of incubation.
Figure 3 shows histological and immunofluorescence data – no data generated from a cell viability assay is included in this figure. The additional sentence included on lines 328-9 (see Reviewer 1, point 1 in the attached document) helps clarify cell viability further.

Round 2
Reviewer 1 Report
The authors have addressed all of my concerns.
Reviewer 2 Report
Authors were adequately responded for all comments.